



# Characterisation of uncertainties in an ocean radiative transfer model for the Black Sea through ensemble simulations

Loïc Macé[12], Luc Vandenbulcke[1], Jean-Michel Brankart[2], Pierre Brasseur[2], and Marilaure Grégoire[1]

[1]FOCUS-MAST research group, Department of Astrophysics, Geophysics and Oceanography, University of Liège, Belgium
[2]Univ. Grenoble Alpes, CNRS, INRAE, IRD, Grenoble INP, IGE, Grenoble, France

**Correspondence:** Loïc Macé (loic.mace@uliege.be)

**Abstract.**

In this paper, we investigate the influence of uncertainties in inherent optical properties on the modelling of radiometric quantities by an ocean radiative transfer model, in particular irradiance and reflectance. The radiative transfer model is coupled to a three-dimensional physical-biogeochemical model of the Black Sea. It describes the vertical propagation of incident irra-

diance within the water column along three streams in downward (direct and diffuse) and upward directions, with a spectral resolution of 25 nm in the visible range. The propagation of the irradiance streams is governed by the inherent optical properties of four major optically active constituents found in seawater and provided by the biogeochemical model: pure water, phytoplankton, non-algal particles and coloured dissolved organic matter. Sea surface reflectance is then derived as the ratio between simulated upward and downward irradiance streams, directly connecting the model with remote-sensed data. In this

configuration, the coupling is in one-way: the radiative transfer model is only projecting model variables into the space of satellite observations, working as an observation operator. In the stochastic version of the model, uncertainties are injected in the form of random perturbations of inherent optical properties of water constituents. Different ensemble configurations are derived and their quality is assessed by comparison with *in situ* and remote-sensed observations.

We find that the modelling of the uncertainties in the radiative model parameterisation allows to simulate distributions of radiative fields that are partially consistent with observations. Ensemble quality is consistent with remote-sensed reflectance data in summer and autumn, especially in the central parts of basin. The quality of the ensemble is lower in winter and early spring, suggesting the existence of another major source of uncertainty, or that the quality of the deterministic solution is insufficient. CDOM dominates absorption in short wavebands with relatively high uncertainty that influences irradiance and

reflectance outputs. This dominant role calls better representation of CDOM to improve model calibration. Contributions from phytoplankton and non-algal particles are more significant for (back-)scattering. The results of this paper suggest that the integration of a radiative transfer model into a physical-biogeochemical model would be beneficial for calibration, validation and data assimilation purposes, offering a better link between model variables and radiometric observations.





## 1 Introduction

The spectral distribution of light in the ocean is closely linked to biological activity, biogeochemical cycles and upper ocean physics (Mobley et al., 2015). The biological productivity of the ocean is directly controlled by the amount of light available to phytoplankton for photosynthesis (Kettle and Merchant, 2008; Gregg and Rousseaux, 2016). The spectral range for photosynthetically active radiation (PAR) corresponds mostly to the visible range and represents about 45% of the total incident radiation at sea surface. Upon reaching the ocean surface, a small part of the incoming radiation is directly reflected depending on the

surface albedo. The rest of the solar irradiance is transmitted to the ocean and its vertical spectral distribution is determined by the optical properties of seawater components such as water molecules, dissolved and particulate materials of various sizes and living material (Mobley et al., 2015). The characterisation of those properties is therefore essential. Marine optics also influences temperature, as most of the absorbed irradiance is converted into energy to heat up the water column (Baird et al., 2020). Those processes participate in modifying the upper ocean stratification and turbulence, thus influencing the distribution

of optically active components close to the ocean surface. As such, a bio-optical feedback exists between phytoplankton, upper ocean physics and surface circulation (Fujii et al., 2007; Skákala et al., 2022; Cahill et al., 2023).

Optically active constituents are defined by their ability to absorb and scatter radiation, called inherent optical properties (IOPs). IOPs depend on the medium and are independent from the ambient light field. They consist in absorption and

(back-)scattering spectra for each water constituent. The determination and parameterisation of IOPs are a major challenge in modelling coupled physical-bio-optical processes (Manizza et al., 2005; Werdell et al., 2018). Most of the vertically resolved biogeochemical models solve only the direct downward component of light in a one-stream model, with a single equation describing the decrease in irradiance with depth following Beer's law (Lengaigne et al., 2006), in a limited number of spectral bands - usually two or three. They usually differentiate the red from the visible part which, in some cases, is separated in the

blue and green wavebands (Aumont et al., 2015; Butenschön et al., 2016; Grégoire et al., 2024). The absorption and (back-)scattering processes are then not distinguished and are merged within a more general attenuation coefficient to represent the attenuation of light over the water column. Over the years, various models were developed with different approaches to refine the representation of light in models (Gregg and Casey, 2009; Mobley et al., 2009).

Remote-sensed optical observations rely on passive radiometers to measure top-of-atmosphere radiance in the visible and near infrared bands. Then, the use of atmospheric corrections allows to estimate water-leaving irradiance to compute remote sensed reflectance ($R_{RS}$) in several wavebands. In units of $sr^{-1}$, $R_{RS}$ is a measure of the water-leaving irradiance normalised by the above water downwelling irradiance. To compare this data with biogeochemical variables, inversion ocean colour algorithms compute intermediate products such as surface chlorophyll from reflectances in selected wavebands. These algorithms

remain rather uncertain, and a more straightforward use of reflectance would mitigate the influence of imperfect inversion algorithms. However, a one-stream optical model does not simulate the upward irradiance and thus cannot provide estimates of sea surface reflectance. Limited use has been made of optical properties so far in data assimilation (Ciavatta et al., 2014;





Jones et al., 2016), which could change with recent hyperspectral sensors such as those of the PRISMA and PACE missions. In recent years, new optical models have emerged to fill this gap and resolve sea surface reflectance. Some of these models resolve three streams of irradiance by considering scattered irradiance in both downward and upward directions, with higher spectral resolution matching the measured wavebands. As a three-stream optical model adds complexity to coupled modelling frameworks, it also becomes necessary to assess the relative uncertainties in the information produced by the model and by satellite sensors.

In this paper, we couple a 3D ocean physical-biogeochemical model implemented in the Black Sea with a radiative transfer (RT) model as previously used in a global setup (Dutkiewicz et al., 2015). The RT model describes the in-water irradiance along the vertical and in three streams: direct downward, diffuse downward and diffuse upward. It solves the spectral wavebands corresponding to those typically used in remote sensing. The penetration of the spectral irradiance is determined by the absorption and scattering properties of the medium that are derived from concentrations of optically active components, in 33 wavelengths. Following literature, four main optically active elements are considered: water molecules, phytoplankton, non-algal particles and coloured dissolved organic matter (CDOM). Based on the simulated irradiance streams, the RT model is used to estimate sea surface reflectance. Then, we combine the model derived reflectances in selected wavebands to obtain an estimation of surface chlorophyll, as it is done in satellite inversion algorithms. With this approach using reflectance-derived chlorophyll, the difference between model and satellite estimated chlorophyll is reduced thanks to the common use of empirical inversion algorithms. Dutkiewicz et al. (2018) presented a proof of concept for the use of a new estimate of chlorophyll (they introduce a proxy for chlorophyll called "derived chlorophyll $a$"), inquiring on the opportunities provided by the use of RT within a coupled biogeochemical framework. They found that this proxy compared better to measurements that the "actual" chlorophyll from their biogeochemical model, with better performances using region-specific methods rather than a global method. In this paper, we use a similar proxy in the Black Sea to evaluate its potential use in this region. We also aim at building up from their work by analysing more thoroughly the inner operation of the RT model with regard to the propagation of uncertainties in such a framework. Here, we focus on the deep sea where BGC-Argo data are available to derive CDOM estimation. Results presented in this paper are mainly valid for the deep region and not for the shelf for which we lack information on CDOM (Grégoire et al., 2023).

By perturbing IOPs of the four selected water constituents, we also aim to use this implementation to assess the intrinsic uncertainty of the RT model. An ensemble simulation strategy is set up to inject perturbations in the IOPs and evaluate their influence on the outputs of the RT model: irradiance and sea surface reflectance. We first assess the effect of uncertainty in the parameterisation of absorption and (back-)scattering for constituents separately. Then, we estimate the combined effect of uncertainties on irradiance and reflectance fields, and evaluate to what extent the modelling of uncertainties can help in providing distributions of surface reflectance that are consistent with observations. To focus the analysis on the RT model, we use it as an observation operator. The RT model takes information from the biogeochemical model without feeding back to the hydrodynamics and biogeochemistry models, and therefore does not influence temperature and primary production.





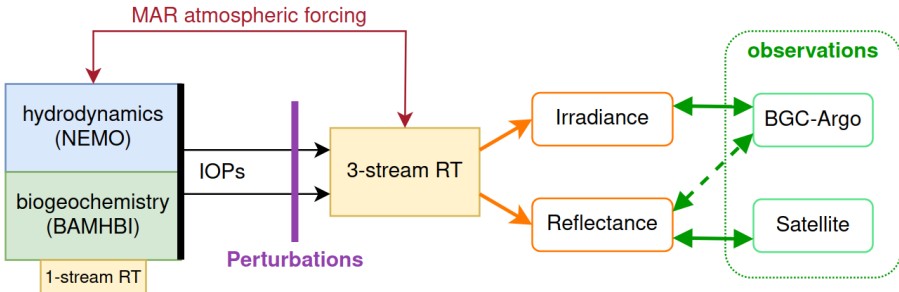

**Figure 1.** Coupled NEMO-BAMHBI modelling framework with the 3-stream RT model functioning in one-way coupling. The RT model is used as an observation operator, projecting model variables into the space of observations. Since it does not feed back to NEMO-BAMHBI, a simple one-stream RT model is kept to compute temperature. Among the outputs of the RT model, irradiance is compared to BGC-Argo data, reflectance is compared to remote-sensed data and BGC-Argo data, through inversion algorithms used to retrieve surface chlorophyll.

The following section presents the deterministic coupled modelling framework and the parameterisation of the optically active components available in the RT model. Section 3 describes the stochastic modelling approach and the introduction of uncertainties within the RT equation. In section 4, we describe the ensemble simulations and results, investigating the impact of uncertainties and the ability of the model to produce relevant surface reflectance distributions. A discussion is finally proposed in section 5, along with possible outlook for future explorations.

## 2 Deterministic modelling framework

In this section, we describe the modelling framework and the observations used for the validation of the system. A schematic description of the modelling framework, the diagnostics provided by the RT model and the relevant observations (detailed further) are presented in Fig. 1. The deterministic model couples a physical-biogeochemical model for the Black Sea with a three-stream RT model. Gathering information from the biogeochemical and hydrodynamics models (section 2.1), the three-stream RT model (subsection 2.2) computes fields of irradiance and sea surface reflectance. These can be compared to both *in situ* and remote-sensed data (section 2.3). The required inputs to perform this operation are provided both by the NEMO-BAMHBI system for IOPs (section 2.4), and forced from MAR atmospheric forcing (section 2.5) for above water irradiance.

### 2.1 Physical-biogeochemical coupled system NEMO-BAMHBI

The physical-biogeochemical coupled system NEMO-BAMHBI is implemented within the Monitoring and Forecasting system of the Black Sea Copernicus Marine Service. The hydrodynamics are solved with NEMO 4.2 and is online coupled with the biogeochemical model. BAMHBI (Biogeochemical Model for Hypoxic and Benthic Influenced areas) is a biogeochemical model that describes the biogeochemical cycles of carbon and nitrogen through the food web going from bacteria up to





mesozooplankton (Grégoire et al., 2008; Grégoire and Soetart, 2010; Grégoire et al., 2024). It explicitly represents processes in the anoxic layer, with a 28-variable pelagic component (including the carbonate system) and a 6-variable benthic component.

The default light penetration scheme used in BAMHBI is a one-stream (i.e. direct downward) RT scheme in three wavebands: two in the visible range and one in the infrared. The sea surface radiation is computed using astronomical forcing to estimate the radiance at the top of the atmosphere and is propagated to the sea surface using the cloud cover data from a regional atmospheric model (MAR) run for the Black Sea. The sea surface radiation is then attenuated with depth following Beer's law with attenuation derived from concentrations of biogeochemical variables: pure water, phytoplankton, suspended
minerals and CDOM. This simple RT model is then used to derive PAR, and the amount of irradiance absorbed is a source term in the equation of conservation of thermic energy in NEMO.

In operational and reanalysis mode, the coupled model works with a horizontal resolution of 2.5 km and 59 unevenly distributed vertical levels, with thinner layers close to the surface and the pycnocline. In this study, we use a horizontal resolution
of 15 km and the same vertical levels. A reduced horizontal resolution is chosen here in order to minimise computational costs, especially in light of ensemble runs that are described in Section 3. In the configuration used in this study, we use a velocity forcing as a boundary condition at the Bosphorus strait for exchanges with the Mediterranean Sea, as described in Stanev and Beckers (1999). It is assumed that there are no exchanges with the Azov Sea. River runoffs and atmospheric deposition of nutrients (phosphorus, nitrate and ammonium) are based on climatology. Initial conditions for the simulations performed in
this study are taken from a longer stable run in a similar model configuration.

## 2.2    The radiative transfer observation operator

The one-dimensional 3-stream RT model is directly inspired from the model described in Dutkiewicz et al. (2015) within the frame of the global MITgcm model. Three streams of irradiance are computed based on absorption and (back-)scattering properties of the medium: a downward direct irradiance ($Ed$) that accounts for transmitted light, a downward scattered irradiance
($Es$) that accounts for light that has been scattered in the forward direction and an upward irradiance ($Eu$) that accounts for the light that has been backscattered towards the ocean surface. Resolving the three irradiance streams allows for comparison with remote-sensed data thanks to the computation of the upward $E_u$ stream at the surface, while the downward $E_d$ and $E_s$ streams are a decomposition of the stream found in one-stream RT models.

The RT model simulates irradiance streams in 33 wavebands ranging between 250 and 4000 nm, with a finer 25 nm resolution in the visible range. This operator is one-dimensional and therefore processes each water column individually. On the vertical, we consider that all scattering happens only either forwards or backwards. We only represent elastic scattering, assuming that inelastic scattering processes are of lower magnitude. Attenuation coefficients are declined into absorption ($a$), scattering ($b$) and backscattering ($b_b$) coefficients, all in units of $m^{-1}$. The propagation of light in the water column is then described by the
following system:





$$\frac{dE_d(\lambda)}{dz} = -\frac{a(\lambda) + b(\lambda) + b_b(\lambda)}{\overline{v_d}} E_d(\lambda) \tag{1}$$

$$\frac{dE_s(\lambda)}{dz} = -\frac{a(\lambda) + r_s b_b(\lambda)}{\overline{v_s}} E_s(\lambda) + \frac{r_u b_b(\lambda)}{\overline{v_u}} E_u(\lambda) + \frac{b(\lambda)}{\overline{v_d}} E_d(\lambda) \tag{2}$$

$$-\frac{dE_u(\lambda)}{dz} = -\frac{a(\lambda) + r_u b_b(\lambda)}{\overline{v_u}} E_u(\lambda) + \frac{r_s b_b(\lambda)}{\overline{v_s}} E_s(\lambda) + \frac{b_b(\lambda)}{\overline{v_d}} E_d(\lambda) \tag{3}$$

where $r_s$ and $r_u$ are dimensionless normalised effective scattering coefficients, and $\overline{v_d}, \overline{v_s}, \overline{v_u}$ are average cosines accounting
for the angle of incidence of light, also dimensionless (Aas, 1987). These coefficients are approximated with constant values
and detailed in Table 1. This system is closed for each water column with two surface conditions provided by the radiative
forcing on $E_d$ and $E_s$, and one bottom condition imposing $E_u$ to be 0. PAR is then derived with:

$$PAR = \int\limits_{400nm}^{700nm} \left[ \frac{E_d(\lambda)}{\overline{v_d}} + \frac{E_s(\lambda)}{\overline{v_s}} + \frac{E_u(\lambda)}{\overline{v_u}} \right] d\lambda \tag{4}$$

We then define sea surface reflectance $R$ in the uppermost layer $0^-$ (below surface) based on the three streams of irradiance
as follows:

$$R(\lambda, 0^-) = \frac{E_u^{0^-}(\lambda)}{E_d^{0^-}(\lambda) + E_s^{0^-}(\lambda)} \tag{5}$$

The model simulated reflectance is not strictly the same quantity as the remote-sensed reflectance $R_{RS}$ which is estimated
above the sea surface. Following Dutkiewicz et al. (2018), two corrections are performed to make the modelled reflectance
comparable to $R_{RS}$ satellite measurements. The subsurface reflectance is first adjusted so that upwelling irradiance stream
$E_u$ is transformed into upwelling radiance as seen by satellite instruments. This transformation is performed according to
bidirectional reflectance distribution function (BRDF) of the ocean surface. The BRDF depends on many variables such as
the wave state at the surface, the solar zenith angle and the optical properties at the air-sea interface (Morel et al., 2002). It is
simplified here in a constant coefficient $Q$. An estimate for Q is generally comprised between 3 and 6 sr. In this study, we set
Q to 4.5 sr.

$$R_{RS}(\lambda, 0^-) = \frac{R(\lambda, 0^-)}{Q} \tag{6}$$

An additional correction is performed to account for interface effects. It aims at translating subsurface remote-sensed re-
flectance to above-surface remote-sensed reflectance (Lee et al., 2002).

$$R_{RS}(\lambda, 0^+) = \frac{0.52 R_{RS}(\lambda, 0^-)}{1 - 1.7 R_{RS}(\lambda, 0^-)} \tag{7}$$

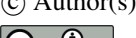



**Table 1.** Average cosines and normalised effective scattering coefficients used in RT equations.

| Parameter | Value | Unit |
|-----------|-------|------|
| $\overline{\upsilon_d}$ | 1.5 | [-] |
| $\overline{\upsilon_s}$ | 0.83 | [-] |
| $\overline{\upsilon_u}$ | 0.4 | [-] |
| $r_s$ | 1.5 | [-] |
| $r_u$ | 3 | [-] |
| $Q$ | 4.5 | 1/sr |

The Copernicus Marine Service provides a multi-satellite product of sea surface reflectance for the Black Sea. It combines
measurements from the Sentinel-3A, Sentinel-3B, Aqua, Suomi NPP and JPSS-1 satellites. This L3 product is available from
1997 and provides daily measurements of sea surface reflectance with a horizontal resolution of 1 km, in 6 wavebands centred
on 412, 443, 490, 510, 555 and 670 nm.

### 2.3 Ocean colour algorithms

Ocean colour algorithms are used to estimate surface chlorophyll from sea surface reflectance. In the Black Sea, the Copernicus
Marine Service uses a merged product between two algorithms: a band-ratio algorithm and a neural network based method
(Zibordi et al., 2015). The neural network is primarily used for complex waters such as coastal areas of the Northwestern
continental shelf of the Black Sea. The blue/green band-ratio algorithm is based on reflectances in the 490 and 555 nm wave-
bands and is used to derive an estimate of surface chlorophyll (Kajiyama et al., 2018). We choose to reproduce the band-ratio
algorithm to derive an estimate of surface chlorophyll based on the reflectances simulated in the RT model (Dutkiewicz et al.,
2018). In the following, we call this reflectance-derived chlorophyll estimate rCHL:

$$log(rCHL) = \sum_{k=0}^{3} c_k \times \left[ log\left( \frac{R_{RS}(490)}{R_{RS}(555)} \right) \right]^k \tag{8}$$

The algorithm described in Kajiyama et al. (2018) provides the $c_k$ coefficients for the Western Black Sea. We choose here to
expand this formulation for the entire basin. This new estimate is not independent from the chlorophyll simulated in BAMHBI
since outputs of BAMHBI intervene in the simulation of reflectances, but provides a quantity that corresponds more closely to
satellite chlorophyll products. From the perspective of modelling, using a reflectance ratio is also interesting as it removes the
uncertainty that arises from the simplification of BRDF.



### 2.4 Optically active components

The absorption and scattering of light by seawater are modelled as the linear combination of individual contributions from pure water, phytoplankton, non-algal particles and CDOM. Sources from literature for specific coefficients used and IOPs to derive each contribution are summarised in Tab. 2. In the following, the subscripts "w", "phy", "prt" and "cdom" denote their respective contributions. We assume here that CDOM only participates in absorption and not in scattering (Dutkiewicz et al., 2015; Álvarez et al., 2023).

$$a(\lambda) = a_w(\lambda) + a_{phy}(\lambda) + a_{prt}(\lambda) + a_{cdom}(\lambda) \tag{9}$$

$$b(\lambda) = b_w(\lambda) + b_{phy}(\lambda) + b_{prt}(\lambda) \tag{10}$$

$$b_b(\lambda) = b_{b.w}(\lambda) + b_{b.phy}(\lambda) + b_{b.prt}(\lambda) \tag{11}$$

#### 2.4.1 Water

The absorption and scattering properties of water ($a_w$, $b_w$ and $b_{b.w}$) are considered well documented by many laboratory ex-
periments (Pope and Fry, 1997; Morel et al., 2007). Water is an important constituent with very high absorbing power in the infrared and ultraviolet, but rather low in the PAR range. Scattering by water is considered isotropic meaning that, in 1D, $b_w = b_{b.w}$. Absorption and scattering spectra for water in the visible range are provided in Fig. 2, with increasing absorption with wavelength and decreasing scattering.

#### 2.4.2 Phytoplankton and non-algal particles

Absorption and (back-)scattering by phytoplankton ($a_{phy}$, $b_{phy}$ and $b_{b.phy}$) are modelled as the sum of individual IOPs for each phytoplankton functional type (PFT) solved in BAMHBI: large flagellates (representative of dinoflagellates), small flagellates (representative of coccolithophores) and diatoms; that are the dominant species in the Black Sea (Silkin et al., 2021). Specific absorption and (back-)scattering coefficients associated to phytoplankton are adapted from Álvarez et al. (2022) to the PFTs
modelled in BAMHBI. Specific coefficients for absorption ($a_{phy}^i$) have units of m$^2$/mgChl while specific coefficients for scattering ($b_{phy}^i$ and $b_{b,phy}^i$) have units of m$^2$/mmolC.

$$a_{phy}(\lambda) = \sum_i a_{phy}^i(\lambda) CHL^i \tag{12}$$

$$b_{phy}(\lambda) = \sum_i b_{phy}^i(\lambda) C_{chl}^i \tag{13}$$

$$b_{b,phy}(\lambda) = \sum_i b_{b,phy}^i(\lambda) C_{chl}^i \tag{14}$$





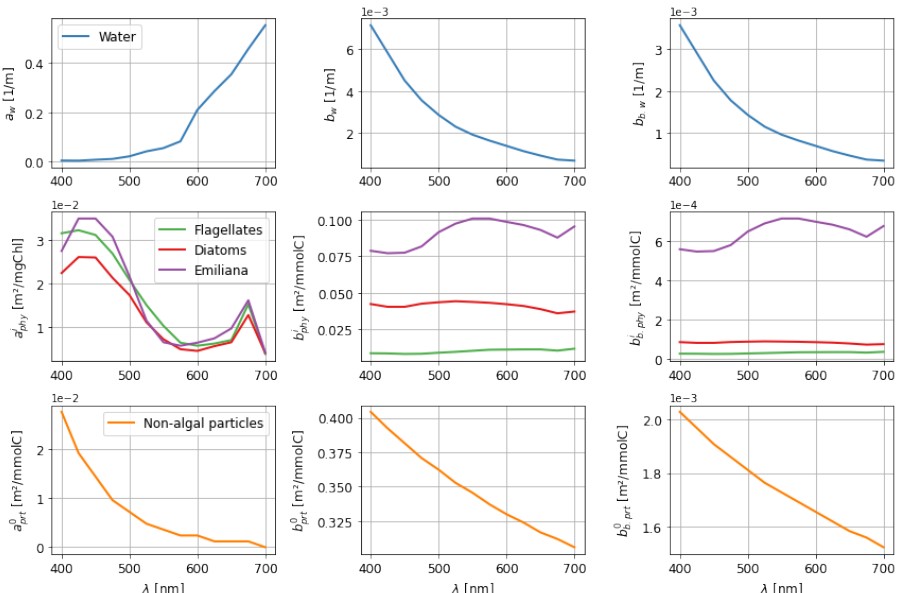

**Figure 2.** IOPs for water ($a_w$, $b_w$ and $b_{b.w}$) (top), phytoplankton ($a_{phy}^i$, $b_{phy}^i$ and $b_{b,phy}^i$) (middle) and non-algal particles ($a_{prt}^0$, $b_{prt}^0$ and $b_{b,prt}^0$) (bottom). Spectral resolution in the model is 25 nm in the visible range.

where $CHL^i$ is the chlorophyll concentration for each PFT in units of mgChl/m$^3$ and $C_{chl}^i$ is the carbon content in each PFT in units of mmolC/m$^3$. Specific absorption and scattering spectra for phytoplankton are provided in Fig. 2.

Similarly, IOPs of non-algal particles ($a_{prt}$, $b_{prt}$ and $b_{b.prt}$) are derived from specific coefficients (respectively $a_{prt}^0$, $b_{prt}^0$ and $b_{b,prt}^0$) from Gallegos et al. (2011) and Álvarez et al. (2022). The computation of these coefficients rely on the use of particulate organic carbon (POC; computed dynamically in BAMHBI) as a proxy for particle concentration, assuming uniformity in the size and distribution of particles as in Dutkiewicz et al. (2015). Specific absorption and scattering spectra for particles are provided in Fig. 2.

$$a_{prt}(\lambda) = a_{prt}^0(\lambda)POC \tag{15}$$

$$b_{prt}(\lambda) = b_{prt}^0(\lambda)POC \tag{16}$$

$$b_{b,prt}(\lambda) = b_{b,prt}^0(\lambda)POC \tag{17}$$

### 2.4.3 CDOM

Although a major contributor to irradiance absorption within the water column, CDOM is not explicitly simulated in the NEMO-BAMHBI framework. We choose here to derive a forcing for $a_{cdom}$ from a collection of BGC-Argo irradiance pro-



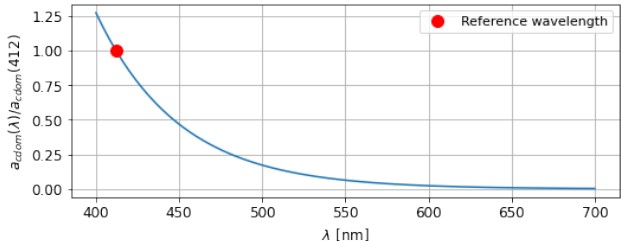

**Figure 3.** Absorption spectrum for CDOM as the ratio with absorption at the reference wavelength $\lambda_{ref} = 412$ nm.

files. The RT model is run by deriving IOPs from BGC-Argo data to reproduce irradiance profile at a reference wavelength $\lambda_{ref} = 412$ nm by optimising $a_{cdom}$ to fit the measured profiles. The description of data, the setup and detail of the method of this experiment are presented in Appendix A.

The forcing for $a_{cdom}(\lambda_{ref})$ depends on time and seawater density $\rho$ as a way of accounting for ocean physics and spatial
and seasonal variability. It also accounts for the increase in CDOM, and therefore in $a_{cdom}$ with depth. This approach is not valid in coastal areas and on the continental shelf, where we observe low densities due to large river discharges of freshwater. With this method, low density areas are associated with low CDOM, which goes against the high biological activity noticed around river estuaries. Since this forcing is generated using data from the central parts of basin, we acknowledge that it will not be representative of shallow areas. We interpolate this dataset as to generate annual cycles for each density scale.


CDOM is dominant in the blue and decreases for longer wavelengths, parametrised with an exponential law for the absorption coefficient of parameter $S_{cdom}$ (Twardowski et al., 2004; Kitidis et al., 2006; Dutkiewicz et al., 2015). A range of values for this slope is prescribed in Álvarez et al. (2022), and we fit it using BGC-Argo irradiance profiles at 380 and 490 nm in the benchmark 1D RT model. We use $S_{cdom} = 0.02$ nm$^{-1}$ for our model, leading to the spectrum in Fig. 3. The uncertainty
associated to this extrapolation coefficient will be studied in the next section of the paper. Our forcing for $a_{cdom}$ is finally written:

$$a_{cdom}(\rho, \lambda) = a_{cdom}(\rho, \lambda_{ref}) e^{-S_{cdom}(\lambda - \lambda_{ref})} \tag{18}$$

### 2.5 Atmospheric forcing

The ocean model is forced with the outputs of a MAR regional atmospheric model configuration (Gallée et al., 2013). Ran over
the Black Sea, MAR provides the boundary conditions for the ocean physics: wind velocity, humidity, precipitation, 2 m temperature, mean sea level pressure. The spectral RT model also requires a boundary condition for spectral irradiance. MAR has been extended with a spectral radiative scheme, ecRad, developed by Hogan and Bozzo (2018) to represent the radiative effect



**Table 2.** Specific IOPs for the main water constituents and their references.

| | | |
|---|---|---|
| $a_w, b_w, b_{b.w}$ | 1/m | Pope and Fry (1997); Morel (1974) |
| $a_{phy}^i$ | m²/mgChl | adapted from Álvarez et al. (2022) |
| $b_{phy}^i, b_{b,phy}^i$ | m²/mmolC | adapted from Álvarez et al. (2022) |
| $a_{prt}^0, b_{prt}^0, b_{b,prt}^0$ | m²/mmolC | Gallegos et al. (2011) |
| $a_{cdom}$ | 1/m | calibrated with BGC-Argo data |
| $S_{cdom}$ | 1/nm | calibrated with BGC-Argo data |

of clouds and aerosols in the ECMWF integrated forecasting system. An accurate description of gas optics is also included in the ecCKD tool used within ecRad (Hogan and Matricardi, 2022). Using this extension MAR provides the direct and scattered

downward irradiance just above the sea surface in the 33 wavebands selected for the ocean RT model. Irradiance below sea surface is then derived by considering surface albedo. The MAR configuration extended with the ecRad scheme is described and validated in Grailet et al. (2024).

In this study, we consider that the surface radiative forcing is accurate, and ignore its uncertainty. Although an error in the sur-

face irradiance would propagate within the water column, the IOPs of water constituents would not be altered. Irradiance fields in the water column would be biased due to surface error, but reflectance would not be significantly influenced as it is a variable that is normalised by the incident light. We would therefore obtain very similar results regardless of the surface forcing. This change also would not influence the rest of the NEMO-BAMHBI system in the current one-way configuration of the coupled model. The scope of this paper is thus focused on the parameterisation of IOPs in the RT model used as an observation operator.


## 3 Modelling of uncertainties

In this section, we describe the method applied to transform the deterministic RT operator into a probabilistic model, simulating explicitly internal sources of uncertainties of the model. Within the RT operator, the IOPs of phytoplankton, non-algal particles and CDOM are uncertain and we aim at quantifying the influence of these uncertainties on the computation of irradiance and

reflectance fields. A generic method is presented in this section to perturb IOPs in a similar way for the 3 constituents.

### 3.1 Stochastic fields

All stochastic perturbations introduced in the model are produced using the generic approach implemented in the NEMO framework by Brankart et al. (2015), and subsequently used in the context of coupled physical-biogeochemical modelling by Garnier (2016) and Popov et al. (2024). We start by generating Gaussian processes, characterised by their mean and time-space



covariance. The stochastic processes are updated at each time step of the model to simulate the evolution of uncertainties in time, and correlated in time with a time correlation representative of biogeochemical processes. In our application, we used first order autoregressive processes (AR1), which means that the value of the process $\eta$ at time step $t_k$ only depends on $\eta$ at the previous time step $t_{k-1}$:

$$\eta(t_k) = \varphi\,\eta(t_{k-1}) + \sqrt{1-\varphi^2}\,w \tag{19}$$

where $w$ is a random Gaussian noise of mean $\mu_0$ and standard deviation $\sigma_0$, and:

$$\varphi = e^{-1/\tau} \tag{20}$$

where $w$ is a normalised white noise (with zero mean and unit standard deviation). With Eq. (19), we can produce AR1 processes with a correlation function given by:

$$corr(\Delta t) = \exp(-\Delta t/\tau) \tag{21}$$

where $\Delta t$ is the time difference and $\tau$ is the decorrelation length scale (both in number of time steps). It represents a decorrelation time at which the influence of a perturbation is $1/e$, and after which it tends to 0. With Eq. (19), we obtain processes with zero mean and unit standard deviation, which can then be rescaled to obtain processes with the required mean ($\mu_0$) and standard deviation ($\sigma_0$).

By applying this method independently at each model grid point, we obtain maps of independent stochastic processes that are uncorrelated in space. Space correlation is obtained by applying a filtering operator on the uncorrelated noise. Here, we use a Laplacian filter to generate a perturbation field $\eta^{'}$ from the above time-correlated point-wise time series of perturbations of the 4 neighbouring grid points.

$$\eta^{'}_{i,j}(t_k) = 0.5\eta_{i,j}(t_k) + 0.125(\eta_{i-1,j}(t_k) + \eta_{i+1,j}(t_k) + \eta_{i,j-1}(t_k) + \eta_{i,j+1}(t_k)) \tag{22}$$

where $(i,j)$ is the index of any grid point. This filtering process can be repeated several times to widen the spatial correlation. In this study, this filtering is repeated five times, thus generating a spatial correlation length scale of 75 km (i.e. five times the spatial resolution of 15 km). The resulting field is normalised again to maintain the required standard deviation.

This transformation produces stochastic fields with space and time correlation structures that remain Gaussian. As we want

to simulate fields of multiplicative noise, we need random positive numbers. Gaussian processes are thus inappropriate and we apply an exponential transform to convert the distribution towards a log-normal distribution, noted $\eta^{''}$.





**Table 3.** Time and spatial correlation properties of the stochastic fields generated to perturb IOPs, with standard deviations of each perturbation.

| Field | Perturbation | Decorrelation time | Laplacian filtering | $\sigma$ |
|---|---|---|---|---|
| $\eta_{phy}$ | Phytoplankton | 30 days | x5 | 0.5 |
| $\eta_{poc}$ | Non-algal particles | 30 days | x5 | 0.5 |
| $\eta_{c1}$ | CDOM absorption at $\lambda_{ref}$ | 30 days | x5 | 0.5 |
| $\eta_{c2}$ | CDOM spectral slope | 30 days | x5 | 0.25 |

$$\eta_{i,j}^{''}(t_k) = e^{\eta_{i,j}^{'}(t_k)} \tag{23}$$

The exponential transform of a Gaussian law transforms the mean and the standard deviation of the initial Gaussian distribution. We want the resulting log-normal distribution to have an unit mean and a standard deviation $\sigma$ as to generate a spread without introducing bias. This constrains the choice for $\mu_0$ and $\sigma_0$ in the initial Gaussian distribution, that are obtained with:

$$\mu_0 = -\frac{\sigma^2}{2} \tag{24}$$

$$\sigma_0^2 = ln(1 + \sigma^2) \tag{25}$$

The resulting two-dimensional maps of stochastic fields follow a log-normal distribution of unit mean and standard deviation $\sigma$. The time-space correlation structure of these AR1 processes can be tuned with the correlation time scale $\tau$ and the number of passes of horizontal Laplacian filter. In the following, we call $\eta$ such a field of perturbation. These stochastic processes are used as multiplicative noises applied to IOPs in the RT model, with parameters as described in Table 3.

### 3.2 Uncertainties on phytoplankton and non-algal particles

Uncertainties in the IOPs are the joint effect of uncertainties in both the concentrations simulated by BAMHBI and the parameterisation of the optical properties (i.e. specific absorption and scattering coefficients) of the optically active components (as in Eq. 12 to 17). Since the modelling of the effect of these two sources of uncertainty on the IOPs can be done in a similar way, we consider that perturbing the abundance of the optically active component mimics the effect of uncertainty in both the abundance and parameterisation. It should be noted that BAMHBI may miss a phytoplankton group that is important in the Black Sea, besides the 3 PFTs it solves, but this uncertainty is not considered here.

Similarly for non-algal particles, we find uncertainties in both the parameterisation of their optical properties and the concentrations. We assume that particles are uniform in size and therefore all share the same specific optical properties. In reality, non-algal particles come in a larger range of sizes and this assumption of uniformity injects uncertainty into the model. Further-





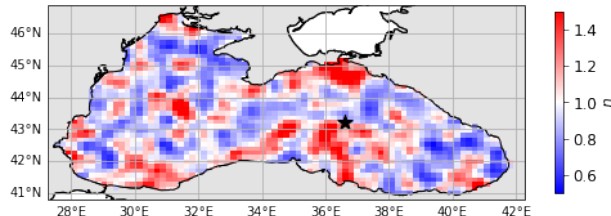

**Figure 4.** Perturbation field generated according to the method described in section 3.1, following a log-normal distribution with unit mean and standard deviation 0.5, with five passes of Laplacian filtering for space correlation. The black star indicates the location of coordinates 43.22N, 36.63E used in the description of results in this paper.

more, the derivation of particle concentration from POC may also be inaccurate. For both chlorophyll and non-algal particles, the perturbations applied are in the form of 2D horizontal fields as described in section 3.1. As such, the perturbation does not

modify the vertical structure of the chlorophyll and POC fields, and cannot account for inaccuracy on the depth of chlorophyll or POC maxima.

We choose to model the uncertainties associated to the model abundance by introducing multiplicative coefficients $\eta_{phy}$ and $\eta_{poc}$ to the computation of absorption and scattering coefficients. The $\eta_{phy}$ and $\eta_{poc}$ fields are generated following the method

described in section 3.1. As such, chlorophyll and POC concentrations are only perturbed at the interface between BAMHBI and the RT model, without influencing the biogeochemical processes described in BAMHBI. Absorption coefficients thus become:

$$a_{phy}^{sto}(\lambda) = \eta_{phy} a_{phy}(\lambda) \tag{26}$$

$$b_{phy}^{sto}(\lambda) = \eta_{phy} b_{phy}(\lambda) \tag{27}$$

$$b_{b,phy}^{sto}(\lambda) = \eta_{phy} b_{b,phy}(\lambda) \tag{28}$$

$$a_{prt}^{sto}(\lambda) = \eta_{poc} a_{prt}(\lambda) \tag{29}$$

$$b_{prt}^{sto}(\lambda) = \eta_{poc} b_{prt}(\lambda) \tag{30}$$

$$b_{b,prt}^{sto}(\lambda) = \eta_{poc} b_{b,prt}(\lambda) \tag{31}$$

$$\tag{32}$$

Decorrelation time is set to 30 days consistently with the scale of biogeochemical changes in the Black Sea. Laplacian filtering generates a spatial correlation length scale of approximately 75 km. Finally, the standard deviation is set here to 0.5, based on the ranges of inherent optical properties found in the literature (Gallegos et al., 2011; Dutkiewicz et al., 2015; Álvarez et al., 2022). An example of the resulting field is shown in Fig. 4. The same perturbation field is used for all 59 vertical layers of the model.




### 3.3 Uncertainty on CDOM

Since CDOM is not explicitly simulated in BAMHBI, its absorption is parametrised from BGC-Argo data. However, this data does not cover the whole basin at all seasons, leading to extrapolations to cover the remaining areas and times. This likely introduces uncertainties in the model, on top of potential observation errors. The approximation of the CDOM absorption spectrum by a decreasing exponential law may also lead to uncertainties in the representation of CDOM in the RT model, as hinted by the several ranges that can be found in the literature for $S_{CDOM}$ (Terzić et al., 2021). Uncertainty on CDOM absorption is modelled in two steps, perturbing separately the reference absorption profile at 412 nm and the exponential slope of the absorption spectrum, both described in section 2.4.2..

We therefore introduce uncertainty on CDOM absorption in two ways. First a $\eta_{c1}$ field of multiplicative factors is used to perturb the reference absorption coefficient from the reference profile. Then, a $\eta_{c2}$ field is used to perturb the exponential slope. Both fields are generated according to the method presented in section 3.1.

$$a_{cdom}^{sto}(\rho, \lambda) = \eta_{c1} a_{cdom}(\rho, \lambda_{ref}) \times e^{-\eta_{c2} S_{cdom}(\lambda - \lambda_{ref})} \tag{33}$$

As for chlorophyll and non-algal particles, the decorrelation time is set to 30 days and the spatial correlation length scale to 75 km. Standard deviation for the perturbation of the reference profile is set to 0.5, based on the ranges of CDOM profiles found in the BGC-Argo observations. This standard deviation accounts for the rather simplified representation of spatial and temporal variability in CDOM concentration and absorption power. Standard deviation for the exponential slope is set to 0.25 based on the range of possible values found in the literature (Terzić et al., 2021). The same perturbation fields is used for all 59 vertical layers of the model.

### 4 Ensemble simulations

In this section, we explore the impact of simulating uncertainties as described in section 3 in the modelling framework complemented by the RT model described in section 2. A deterministic reference simulation and 4 distinct ensembles are simulated. E1 and E2 each are 10-member ensembles with perturbations on the optical properties of phytoplankton only and non-algal particles only, respectively, as described in section 3.2. E3 is a 10-member ensemble with perturbations on the optical properties of CDOM only, as described in section 3.3. Both the CDOM absorption reference profile at $\lambda_{ref}$ and the spectral dependency of CDOM absorption are perturbed in this ensemble. E4 is a 20-member ensemble combining the perturbations used in E1, E2 and E3 (i.e. on chlorophyll, non-algal particles and CDOM).

All ensemble simulations are performed over 15 months: a spin-up time of 3 months between October and December 2016 followed by the simulation of 2017. The simulated domain covers the entire Black Sea. 10 members are simulated for ensembles E1, E2 and E3 as few members are deemed enough to evaluate the influence of each perturbation. The most important





results described in this section come from ensemble E4, combining perturbations. As such, 20 members were simulated to allow a more reliable statistical analysis of the ensemble. However, when E4 is directly compared to E1, E2 or E3, it is limited to its first 10 members for consistency.

We begin by studying the individual influence of perturbations on spectral irradiance profiles and surface reflectance. Then, we assess the impact of combination of uncertainties in E4. Finally, we compare ensemble results to observation data to evaluate the ability of the different ensembles to produce distributions that are consistent with observations.

## 4.1    Influence of perturbations on radiative transfer

The perturbation of IOPs influences the vertical propagation of irradiance in the water column, by modifying the absorption
and (back-)scattering properties of the medium. In the case of CDOM, the perturbation only modifies absorption (see Eq. 1, 2 and 3). Modifying the absorption affects the total amount of irradiance propagating in the water column while a modification of the (back-)scattering coefficients affects the direction of light (downward or upward). The three irradiance streams $E_d$, $E_s$ and $E_u$ are affected by absorption and backscattering. Perturbation of scattering coefficients changes the amount of direct irradiance that is scattered, but does not influence the backscattered stream nor reflectance. It also has no influence on the total
downwelling irradiance defined as the sum of direct and scattered streams.

The dominant optical constituent for absorption and (back-)scattering varies during the season, region and waveband. In the following we focus on the spectral bands used in satellite inversion algorithms in the Black Sea to compute surface chlorophyll: 490 and 555 nm. Appendix B. provides an overview of the model performance in other wavebands in the visible range. Fig. 5
presents time series of perturbed IOPs at 490 nm in ensembles with individual perturbations throughout 2017, in the Eastern part of the basin and at shallow depth (10.6 m). For absorption, we note that the spread around CDOM is the largest, in agreement with the high uncertainty set for CDOM IOPs that are not explicitly modelled in BAMHBI. Scattering coefficients are 10 to 100 times larger than absorption coefficients while backscattering is about 10 times lower. Fig. 7 below illustrates how the perturbation of these coefficients influence the irradiance streams in E4, highlighting that most of the light is converted into the
downward diffuse part $E_s$.

In the Eastern gyre, CDOM has no clear seasonality. It dominates absorption throughout the year with lower values at the end of summer and fall. (Back-)scattering is mainly dominated by non-algal particles except at the end of the year when phytoplankton reaches a similar and even larger contribution. Overall, we identify three different regimes throughout the year. In
spring, we observe high biological activity associated with a phytoplankton bloom. Absorption is mostly dominated by CDOM, with high phytoplankton contribution, while scattering is dominated by non-algal particles, as their concentration increases following the phytoplankton bloom. During summer, we observe a regime of low biological activity once the high concentrations of phytoplankton and particles start to decrease after the bloom. At this stage, CDOM dominates absorption while backscattering is dominated by non-algal particles. During autumn, contribution from phytoplankton increases again until reaching a



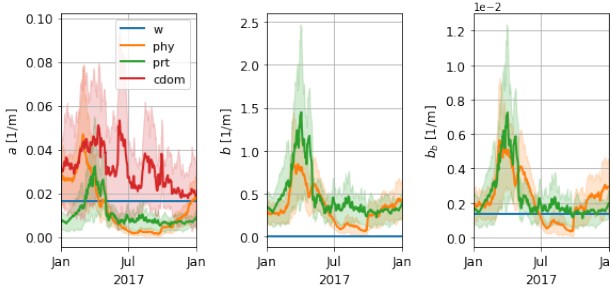

**Figure 5.** Time series of absorption, scattering and backscattering coefficients at 490 nm from the perturbed runs at 43.22N, 36.63E (see location on Fig. 4) in 2017, at 10.6 m depth. The solid lines represents the ensemble mean and the shaded areas covers one standard deviation around the ensemble mean.

level similar to that of CDOM and water for absorption. For scattering, both contributions from phytoplankton and particles gradually increase in autumn. It is worth noting that the introduction of perturbations in this setup allows for a change of the constituent dominating absorption or (back-)scattering. For instance during blooms, perturbing the model may create a switch from a domination of absorption by CDOM to a domination by phytoplankton, or conversely, when contributions are of similar magnitude in the deterministic simulation. These three regimes are valid for the deep sea. In coastal and shallow areas such

as the Northwestern shelf, biological activity follows different patterns. There, contributions of phytoplankton and particles remain high during the whole year, with a lower contribution of CDOM on absorption. On the shelf, scattering and backscattering are mostly dominated by non-algal particles. In this paper however, we focus on the deep central areas of the basin as it is where the CDOM forcing is the most reliable.

Figure 6 presents the seasonal cycle of the surface reflectance $R_{RS}(490)$ and rCHL (as defined in equation 8) generated by the ensembles. Perturbations in the backscattering and absorption influence the upward diffuse irradiance (see Fig. 7a) that in turn modifies surface reflectance. While backscattering directly influences the fractions of light scattered upward in the $E_u$ stream, absorption also has a significant importance as it drives the ability of the water column to propagate irradiance streams. For instance with low absorption, the thickness of the surface layer in which backscattered light eventually reaches back to the

surface increases. In spring, perturbations of chlorophyll and CDOM generate the largest dispersion (Fig. 6a and c). Low biological activity in summer leads to reduced dispersion in ensembles E1, E2 and E3, until in increases again in fall and triggers dispersion of the ensemble members. Although perturbing the IOPs of non-algal particles seems to have a limited influence on rCHL, as in Fig. 6b, the perturbations remain significant in their influence on surface reflectance.

While CDOM shows a large spread at 490 nm, it has little influence on reflectance at 555 nm, as the contribution to absorption is lower at longer wavelengths. On the other hand, the perturbation of contributions from phytoplankton and non-algal particles has a greater influence on reflectance at 555 nm. We also notice that some members drift from the deterministic run or from





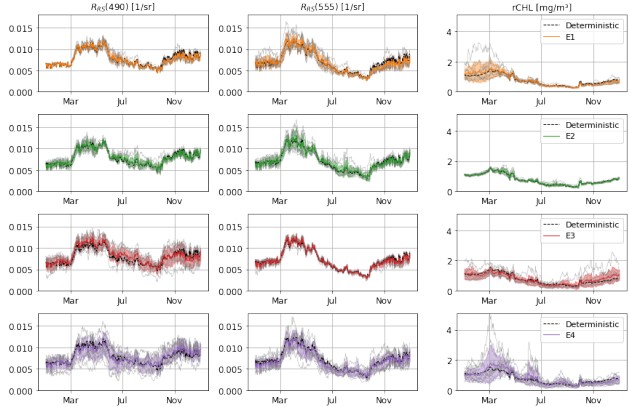

**Figure 6.** Time series of sea surface reflectance at 490 nm ($R_{RS}(490)$) and reflectance derived chlorophyll (rCHL) over 2017 at the location 43.22N, 36.63E. For each ensemble, the standard deviation is represented with a shaded area. Phytoplankton, non-algal particles and CDOM IOPs are respectively perturbed in E1 (in orange), E2 (in green) and E3 (in red). All IOPs except pure water are perturbed in E4 (in purple).

ensemble mean. For instance in the E1 ensemble (perturbation of phytoplankton IOPs), one member (in grey) reaches high values for rCHL during the diatom bloom while reflectance values at 490 nm remain rather close to that of the other members. In this case, it is associated to higher absorption at 555 nm that lowers the backscattered signal. In all experiments, ensemble means remain rather close to the deterministic run, showing that no significant bias is introduced with the ensemble.

### 4.2 Combined vs individual perturbations

The different regimes evidenced with individual perturbations are also highlighted in the E4 ensemble with combined perturbations. The evolution of reflectance and rCHL (Fig. 6d) simulated by the E4 ensemble exhibits similar patterns as those observed in the E1, E2 and E3 ensembles. For surface reflectance, early in the year, the spread of the E4 ensemble is only slightly larger than that of the E3 ensemble (Fig. 6c) at 490 nm in which only CDOM optical properties are perturbed. During the spring bloom, the spread in E4 becomes larger resulting from a combined effect of CDOM (E3), phytoplankton (E1) and non-algal particles (E2). Similarly in summer, and despite the low dispersion observed in E1 and E2, the spread in E4 is greater than in individual perturbations suggesting the influence of non-linearities. CDOM contributes the most to the uncertainty of $R_{RS}$ at 490 nm which is consistent with its higher absorption power in shorter wavelengths. At 555 nm, CDOM contributes less to absorption with higher influence of phytoplankton and non-algal particles. The seasonal evolution of rCHL (right-hand side panels of Fig. 6) simulated by E4 suggests a dominance of phytoplankton early in the year with lower contribution of CDOM the ensemble spread. The extent of the spread is much higher in E4 during the spring bloom than in E1 or E3, exhibiting again the non linear character of the perturbations and amplification by the model non-linear dynamics. In both E1 and E4, one member of each ensemble shows very high values of rCHL compared to the ensemble mean, thus largely increasing the spread of the ensemble. In summer, perturbations in CDOM and phytoplankton dominate the E4 spread with a lower influence





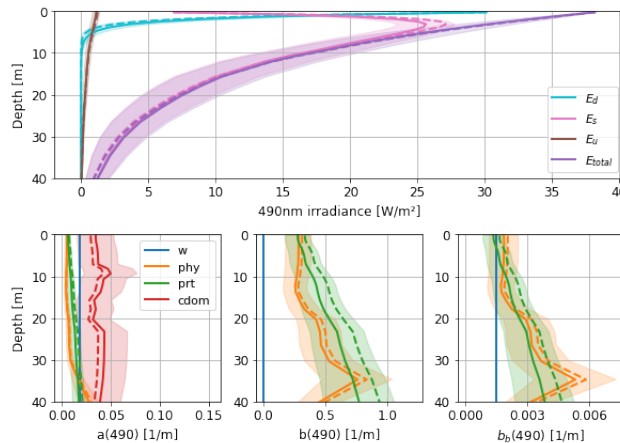

**Figure 7.** Irradiance (top) and IOP (bottom) profiles at 490 nm. Profiles are taken here at 43.22N, 36.63E on 23rd June 2017, at solar zenith. Solid lines represent the ensemble mean with shaded areas representing standard deviation of the ensemble. Dotted lines represent the deterministic run.

of non-algal particles. In autumn, CDOM perturbations dominate again the spread of rCHL.

Fig. 7 shows the effect of combined perturbations of absorption and (back-)scattering coefficients on the three streams of
irradiance in the eastern gyre of the Black Sea in early summer. The influence of perturbations is represented with both the
ensemble mean in dotted lines and with the standard deviation of the ensemble. This figure shows that CDOM dominates
absorption at depth at 490 nm, consistently with the results presented in Fig. 5. (Back-)scattering gradually increase with depth
and is dominated by phytoplankton and particles that result from a former bloom, with also a significant contribution of water
to backscattering at surface. Below 2-3m the downward diffusion stream largely dominates the other two streams and pen-
etrates waters below 40 m. The direct downward stream $E_d$ at 490 nm is fully absorbed after 5 to 7 m, mostly by CDOM,
and (back-)scattered by phytoplankton and non-algal particles. The upward backscattered stream $E_u$, that contributes to the
reflectance, is the smallest one and is present up to around 30 m.

Most of the simulated variability in the total stream $E_{total}$ occurs at low depths and mainly results from variability in the
scattered stream $E_s$. The impact of the uncertainty in IOPs on the direct and backscattered irradiances ($E_d$ and $E_u$) is lower
in magnitude due to lower absorption and backscattering coefficients. In this example, the combined effect of perturbations
results in an increase of the total absorption (i.e. the ensemble mean is larger than the deterministic solution over the whole
column) while for (back-)scattering it is opposite. This results in a lower ensemble mean irradiance compared to the deter-
ministic simulation close to the surface, while the backscattered $E_u$ stream is also lower (Fig. 7a). The uncertainty on the
total light increases with depth and below 20 m has a spread comparable to the average light. At depth, this range is consistent





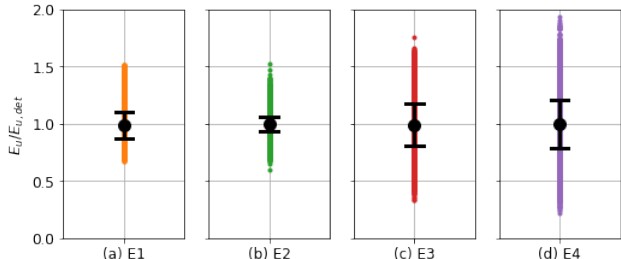

**Figure 8.** $E_u$ at 490 nm distribution at surface normalised by its deterministic value, for the whole basin on 23rd June 2017 and each experiment. Circles and black lines represent the ensemble mean and standard deviation.

with higher chlorophyll-a dynamics. For comparison, standard deviation on $E_{total}$ is 21% at 10 m depth and 53% at 30 m depth.

Figure 8 presents the spread of the upward diffuse irradiance $E_u$ at 490 nm at the surface simulated over the whole Black Sea in summer in the 4 ensemble experiments. Since the incident irradiance is the same in all ensembles, the difference in
$E_u$ results only from the perturbation of IOPs. It shows that CDOM perturbation has the largest influence on $E_u$, which subsequently influence surface reflectance. While CDOM does not backscatter light, it absorbs the three streams including $E_u$ (Eq. 3). Perturbation of phytoplankton IOPs has the second largest influence with non-algal particles having the least influence. Their contribution is two-fold: directly on backscattering and through absorption. Standard deviations and ensemble means are summarised in Table 4. It is worth noting that the ensemble means are kept close to 1, showing that no bias is introduced with
the perturbations.

The effect of combined perturbations in E4 is shown in Fig. 8d. E4 shows greater extrema in the ensemble than E1, E2 and E3, but the standard deviation of $E_u$ remains rather close to that of E3 where only the CDOM was perturbed, with 20.7% for E4. The amplitude of the combined perturbation is therefore lower than the sum of amplitudes of individual perturbations,
exhibiting the non-linearity of this relationship. The increase in spread still shows that perturbations are amplified when combined and do not cancel out each other. It is also worth noting that the standard deviation with each ensemble is lower than the 50% standard deviation used to generate perturbations.

| $E_u$ | at 490 nm Ensemble mean | Standard deviation |
|---|---|---|
| E1 | 0.985 | 11.4% |
| E2 | 0.993 | 6.5% |
| E3 | 1.013 | 18.8% |
| E4 | 0.996 | 20.7% |

**Table 4.** Ensemble statistics for normalised upward irradiance fields $E_u/E_u^{\text{ref}}$



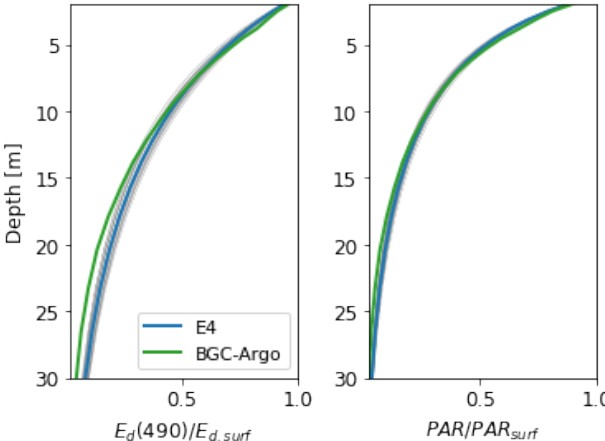

**Figure 9.** Profiles of downwelling irradiance at 490 nm and PAR from BGC-Argo data and the E4 ensemble normalised by their surface values. Each thinner grey line represents an individual member of the ensemble.

## 4.3 Comparison to observations

In this subsection, we assess the quality of the E4 ensemble by comparing the distribution of the simulated irradiance and reflectance fields with observations.

### 4.3.1 BGC-ARGO profiles

A collection of 108 BGC-Argo profiles from floats 6901866 and 7900591, collected over the year 2017 in the Eastern gyre of the Black Sea, is used to compare the ensemble E4 to *in situ* observations. As we want to assess the representation of marine
optics within the water column regardless of the surface forcing, we normalise irradiance profiles by the surface values. Fig. 9 shows the average normalised profiles of simulated and measured downwelling irradiance at 490 nm and PAR. It shows that while there is a good agreement close to the surface for both variables, divergences appear at depths greater than 15 m. This result, while expected since the calibration of CDOM optics was performed on the same type of data, illustrates that the RT model is able to represent the total amount of light in a way that is consistent with *in situ* data at shallow depths. Below 15 m,
when the divergence start to occur, about 70% of the irradiance has already been absorbed.

Comparison to BGC-Argo measurements reveal a rather strong agreement between observed irradiance profiles and their equivalent computed by the RT model, although not all the observed profiles fall within the ensemble spread throughout the year. Fig. 10 shows the time series of $E_d(490)$ and PAR at 10.6 m depth, both directly and normalised by their surface values.
While the observations remain contained within the ensemble spread during winter and spring, the ensemble then overestimates the irradiance and PAR (i.e. does not absorb enough) during early summer. It is worth noting that some residual error can likely be attributed to the BGC-Argo measurements (e.g. sensor drift, sensitivity to temperature) and explain some of





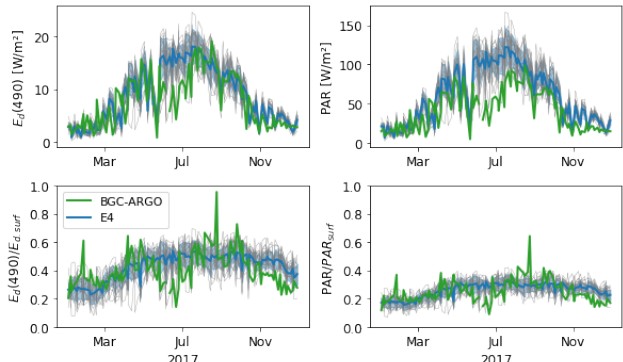

**Figure 10.** Time series over 2017 of downwelling irradiance (i.e. $E_d + E_s$) at 490 nm (top left) and PAR (top right) at 10.6 m depth for the E4 ensemble and BGC-Argo data from floats 6901866 and 7900591. They are also shown normalised with their surface value (bottom). Floats here both drift in the Eastern part of the basin during this period. Each thin grey line represents an ensemble member. The standard deviation of the ensemble is represented with the shaded blue area.

the discrepancies observed in this comparison. The agreement is then better during early autumn, still with an overestimation in the final months of the year. This appears more clearly in the normalised profiles where the biases appear distinctly in summer and at the end of the year. We also notice some outliers in the observations that seem to indicate isolated events of underestimated irradiance, but those data points may indicate local fine-scale variations that are not captured, or observation error.

While part of the data remains outside of the ensemble spread, it seems that the ensemble is able to capture most of the measurements from BGC-Argo data. Fig. 11 shows a rank histogram gathering the normalised data from the time series in Fig. 10 illustrating the dispersion of observation data within the ensemble members. A rank histogram (Candille and Talagrand, 2004) aims at assessing the reliability of an ensemble. It sorts observations within the ensemble of corresponding simulated data. A flat histogram evidences perfect reliability, i.e. an ensemble distribution that matches the distribution of observations. Each observation is ranked within the sorted ensemble, with the extreme ranks corresponding to observations that are lower or higher than all realisations of the ensemble. To compute this histogram, we consider an observation error of 6% on the normalised irradiance measured by the BGC-Argo floats. This value is that prescribed for the satellite products (information from manufacturer - https://biogeochemical-Argo.org/measured-variables-general-context.php).

A large number of observations are not captured by the ensemble (Fig. 11), suggesting an under-dispersed ensemble. At lower depths, Fig. 9 showed that the ensemble overestimates irradiance and PAR regardless of the perturbation. The ensemble therefore does not seem able to fully capture the measured irradiance streams. It is worth mentioning that the amount of observations may not be sufficient here to provide a full picture of the reliability of the ensemble. While it provides an overview of the ability of the model to reproduce distribution of downwelling irradiance, the greater amount of observations provided by satellite data offers a bigger dataset to assess the ensemble skill.



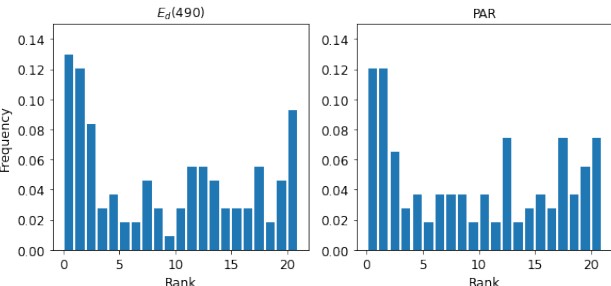

**Figure 11.** Rank histograms using BGC-Argo data in 2017 for $E_d(490$ (left) and PAR (right), at 10.6 m depth. Observations are sorted within the E4 ensemble.

### 4.3.2 Remote-sensed data

Comparison with remote-sensed data allows a better coverage of the basin both spatially and temporally. We start by studying the temporal signature from the ensemble by comparing time series of reflectance in the Eastern gyre at 490 and 555 nm, and rCHL. Fig. 12 shows that the simulated reflectances are higher than observations for most of the year at the exception of the summer period where higher reflectances are sensed by satellite instruments. These correspond most likely to a coccolitophore bloom (Kubryakov et al., 2021), whose backscattering magnitude is largely underestimated in the model. This underestimation

of reflectance by the model is observed for both the 490 and 555 nm wavebands. Interestingly the chlorophyll from remote-sensed reflectance does not peak during the coccolithophore bloom event. The bias is greatly reduced in the estimation of rCHL. Indeed as this quantity is a ratio between reflectances in two wavebands that are rather close, uncertainties on measurement and corrections such as the BRDF coefficient partly cancel out. However, we still observe a clear overestimation by the model during winter and early spring. From June and until the end of the year, observations are captured by the ensemble at

this location of the eastern gyre.

To better characterise the quality of the ensemble of rCHL outputs, we compute the rank histograms over the whole basin at three dates selected in winter (March 5th), end of spring (June 8th) and end of summer (September 13th) during days with limited cloud cover for better satellite coverage over the whole basin. The observation error used to generate those rank his-

tograms is set to 30%, as in Popov et al. (2024). Fig. 13 presents the resulting rank histograms and the associated maps. Rank maps show the spatial distribution of observation ranks. In this representation, we discard coastal and shallow areas as these are more complex waters where the ocean colour algorithm described in section 2.5 shows weak performances. Then, only areas of the basin with a depth greater than 150 m are considered.

Fig. 13 shows the spatial and temporal localisation of the bias in rCHL. It confirms the presence of the bias in the beginning of the year as on March 5th, with only few observations that are captured by the ensemble in the South of the Black Sea. In June, the bias seems to decrease with a greater amount of observations that are captured by the ensemble in the Eastern part





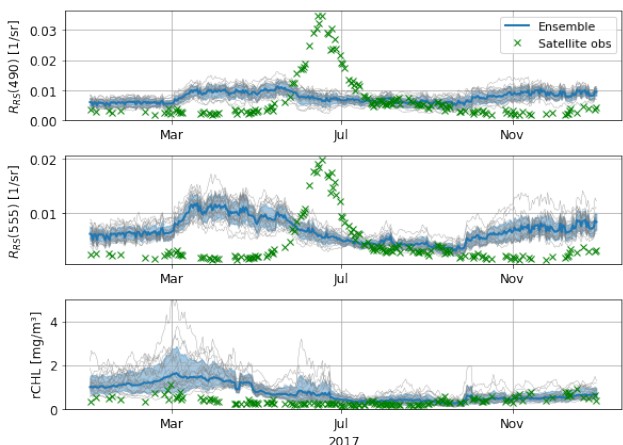

**Figure 12.** Time series over 2017 of $R_{RS}$ at 490 (top) and 555 nm (centre), and reflectance-derived chlorophyll (bottom). Series are taken in the eastern gyre (43.22N, 36.63E). The standard deviation of the ensemble is represented in shaded blue and each thin grey line corresponds to a member of the ensemble.

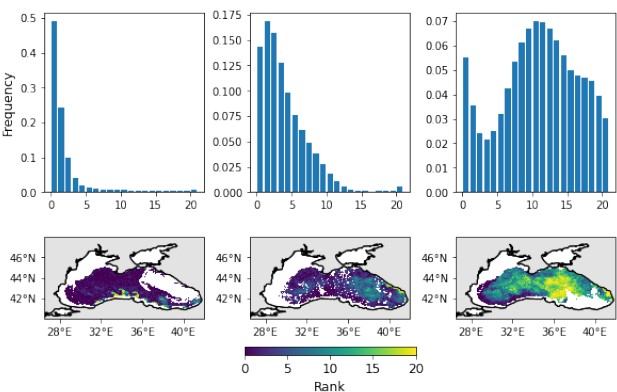

**Figure 13.** Rank histograms and maps for rCHL on 5th March (left), 8th June (centre) and 13th September (right).

of the basin. This is confirmed on the rank histogram that presents a lower peak and better distributed data. The data remains however gathered on the left side of the rank histogram, exhibiting an overestimation of rCHL by the ensemble. The bias there seems mainly located at the periphery of the gyres, while observations close to the centre of both gyres appear to be better captured. In September, the flatter rank histogram shows that the model captures the observations in a more consistent way. We may still observe a positive bias in the Southwestern parts of the basin, but the central areas are consistently represented.

Finally, we proceed to a more complete comparison between the chlorophyll estimated by the satellite and BGC-Argo with that predicted by the BAMHBI model and estimated from the reflectance rCHL. We subsample model and satellite surface chlorophyll data at the times and locations of BGC-Argo measurements. For the BGC-Argo and BAMHBI deterministic run,





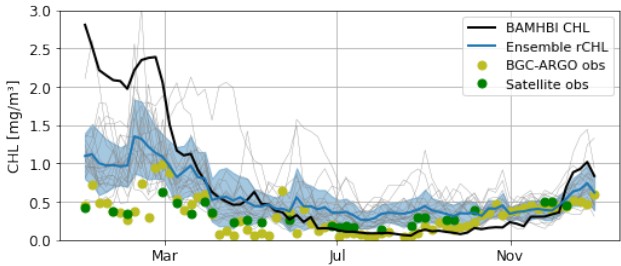

**Figure 14.** Surface chlorophyll estimated by BGC-Argo, satellite and the model (dynamically simulated by BAMHBI) and estimated from reflectance (rCHL). Comparisons are shown at times and locations of BGC-Argo measurements in 2017, along Argo float 6901866.

surface chlorophyll is defined as the average concentration over the top 10 m. The resulting time series are presented in Fig. 14.

Despite the bias observed in rCHL in the first months of the year, surface chlorophyll derived from reflectance appears to be closer to observations than the chlorophyll computed by the biogeochemical model. In winter, both estimates of chlorophyll overestimate its concentration, but the deviation is lower with rCHL. Both methods then provide similar estimates in spring before exhibiting contrasting results in summer and autumn with a slight overestimation for rCHL and underestimation for BAMHBI chlorophyll. In the last months of the year, both estimates remain close to observations. Despite the use of BAMHBI chlorophyll as an optical component that governs the light propagation and then the surface reflectance from which is derived rCHL, both signals do not seem to always follow similar patterns throughout the year, even during blooms. This highlights the role of the other constituents when biological activity is high. We also note that *in situ* BGC-Argo data is rather consistent with the remote-sensed data.

## 5   Discussion

In this section, we first analyse the quality of the generated ensemble to represent the variability in the observations. We then summarise the main impacts of IOPs uncertainties on the simulated radiative fields. Limitations of the approach are discussed and the potential of using radiometric data to better constrain biogeochemical models is discussed based on our results.

### 5.1   Quality of the ensemble

By taking uncertainties into account in the formulation of the RT model, we produce ensemble distributions of irradiance and rCHL that provide enriched information with respect to a single deterministic simulation. To test the quality of the ensembles, we assess the consistency of the simulated distribution of irradiance and reflectance-derived chlorophyll (rCHL) with remote-sensed and *in situ* observations. The ensemble E4 that combines all the perturbations under-disperses the distribution of the downwelling irradiance at 490 nm and PAR when compared to BGC-Argo data, with an increasing bias observed with depth.





From the analysis of surface reflectance distributions, we notice that the consistency of the ensemble generated rCHL with
satellite data varies across the year. The ensemble shows good agreement from the end of spring until late autumn in most parts
of the basin (excluding shallow areas such as the continental shelf). Rank histograms (Fig. 13) evidence that the distribution of
rCHL is closer to that of observations in the second half of the year. In winter and early spring, the consistency of the ensemble
data with observations is lower and observations are not captured. This is indeed the period during which the deterministic
biogeochemical model strongly overestimates chlorophyll as measured by satellites. Further calibration of the biogeochemical
model or better parameterisation of its uncertainties, including from other sources that were not accounted for, is needed in
those conditions to first improve the deterministic solution.

It should be noted that this analysis is essentially valid in the surface layer and the visible spectral range. Comparison to
BGC-Argo profiles show a mismatch at depth over 15 m (Fig. 9), where about 30% of the incident irradiance is still propa-
gating downward. There, the current parameterisation of absorption may be lacking to accurately represent irradiance profiles.
The uniformity of perturbations applied on the vertical also does not allow to account for an inaccurate depth of phytoplankton
maximum for instance, that has a significant influence on the propagation of irradiance in lower depths. Results obtained at
surface remain close to observations and confirm the benefits of using ensemble methods to describe uncertain processes and
parameters such as IOPs.


### 5.2   Influence of uncertainties in IOPs on radiative fields

As shown in the times series in Fig. 5, CDOM has the greatest influence on absorption in the blue end of the visible spectrum,
and less so at 555 nm. Phytoplankton and non-algal particles dominate scattering and backscattering with seasonal variations
in their relative dominance imprinted by the bloom dynamics. With depth, all contributions increase as concentrations of phy-
toplankton, non-algal particles and CDOM gradually increase. The direct irradiance stream $E_d$ only penetrates a few metres
because of high scattering properties of water constituents, as seen in Fig. 7. Below, the light field is strongly dominated by the
downward diffuse stream $E_s$. The upward stream $E_u$ is always of smaller magnitude and almost null below 25 m. The spread
in the vertical light field is mostly due to the downward diffuse stream. As a consequence of uncertainties in the absorption and
(back-)scattering, its spread increases with depth and below 10 m has a magnitude that can impact photosynthesis and then the
amplitude of the deep chlorophyll maximum. This effect is however not tested here since the light generated by the RT model
is not used to force the biogeochemical model.

As presented in Fig. 7 with $a_{cdom}$ for instance, we sometimes observe a mean effect that deviates the ensemble mean from
the deterministic results. This is likely caused by model non-linearities such as the $S_{cdom}$ coefficient in the formulation of
$a_{cdom}$, despite the introduction of perturbations that do not create bias in the perturbed IOPs. It could also be noted that this
effect could be mitigated by increasing the ensemble size, that could explain deviations in (back-)scattering coefficients. Over-
all, combining perturbations increases the spread of the ensemble compared to single perturbations as exhibited in Fig. 8.



The non-linear character of the model however leads to lower spreads in E4 than the simple sum of spreads with individual perturbations. In all 4 ensembles, spreads in irradiance and reflectance remain lower than the 50% standard deviations used to

perturb the IOPs.

## 5.3 Importance of CDOM

CDOM largely dominates the absorption at short wavelengths (lower than 500 nm) and strongly influences the propagation of the three light streams. Since CDOM is not explicitly simulated by the biogeochemical model, its uncertainty is set as the

largest. The comparison of ensembles shows that the ensemble that combines the different sources of perturbations has similar patterns to that obtained with CDOM perturbations, only with a wider spread. The high influence of CDOM perturbation on outputs of the RT model shows how important this component is and what variations an inaccurate estimation can produce. Perturbing CDOM strongly impact $E_u$ and hence $R_{RS}$. However, most biogeochemical models either strongly oversimplify or do not explicitly solve CDOM dynamics. The reasons are the lack of knowledge and data to parametrise CDOM in the Black

Sea, although its inclusion in models was proven relevant (Terzić et al., 2019). In our study, CDOM is represented as a forcing derived from BGC-Argo data. This approach has limitations since BGC-Argo floats do not cover the shallow areas of the basin. In order to link the CDOM seasonal and spatial variability associated to the physics, the CDOM forcing is scaled according to seawater density, with CDOM being parametrised as an increasing function of density as described in section 2.4.3. However this procedure results in low CDOM values in low density areas which does not match the reality of coastal areas largely influ-

enced by river discharges. Although the perturbation of CDOM absorption accounts for the uncertainty related to derivation of the forcing, the large variations that could be observed in coastal areas are likely not captured by the ensemble because of the different biogeochemical mechanisms at play in those regions. Those results highlight the need for representation of CDOM in biogeochemical models in order to successfully model RT in short wavelengths and provide a more complete assessment of the variability of CDOM IOPs both spatially and temporally.


## 5.4 Limitations of the approach

The optically active components considered here are the three phytoplankton functional groups, non-algal particles and CDOM. Results of this study are influenced by the way these components are modelled in the biogeochemical model. First, as described above, the forcing used to represent CDOM absorption has its limitations, especially close to shallow areas. We also ignore sus-

pended minerals that can also be optically important, limiting the accuracy of our study on the continental shelf (Stramski et al., 2016). Moreover, as in Dutkiewicz et al. (2015), the size of particles is assumed to be uniform which is an oversimplification. The modelling of phytoplankton optics is also largely influenced by the PFTs resolved in the mode, that are representative of diatoms, dinoflagellates and small flagellates. This last group mainly integrates coccolithophores, although BAMHBI does not consider calcification. The analysis of reflectance fields clearly shows that the model misses the reflectance peaks in early June

associated to the coccolithophore bloom. During this period BAMHBI simulates a bloom of small flagellates but its optical



signature is underestimated. This arises both from a flawed parameterisation of IOPs and from an underestimation of the bloom in BAMHBI. We could also investigate the inclusion of more PFTs that are relevant for the Black Sea, such as diazotrophs (Dechenne, 2023) or synechococcus (Uysal, 2001).

There are nonetheless still improvements to be done in the current formulation of the stochastic RT model based on the current variables provided by BAMHBI. Several challenges remain to calibrate properly a deterministic RT model due to the rather limited amount of optical *in situ* data in the Black Sea, as well as of phytoplankton composition. The addition of stochasticity as described in this paper only allows to account for model uncertainty up to a certain extent. For instance, the perturbations applied here do not influence the depth of optically active constituents that would likely need to be perturbed based on the

discrepancies with observations. With the current method, a phytoplankton bloom that is too deep in the model will not see its depth perturbed. In this study, we also chose to only perturb the RT model with no feedback towards the hydrodynamical and biogeochemical models (i.e. the modified light field is not that used in the coupled model). By considering it and other sources of uncertainty (e.g. surface radiation) that affect biological processes, we may be able to capture some of the observations that remain outside of the ensemble E4 spread and provide a more complete description of the sensitivity of the RT model.


## 5.5 Radiometric observations to constrain coupled physical-biogeochemical models

The analysis performed here with this RT model also paves the way to use radiometric quantities for calibration, validation and assimilation in a physical-biogeochemical model in the Black Sea. Comparison to reflectance data still shows high biases that would require prior corrections but the definition of reflectance ratios, or associated variables such as reflectance-derived

chlorophyll (rCHL), shows promising results. Using rather close wavebands, ratios remove some of the uncertainties associated to corrections (e.g. atmospheric, at sea surface).

    This study also provides a better understanding of the strengths and limitations of ocean colour algorithms. While those algorithms aim at estimating surface chlorophyll concentrations, the ocean colour products fundamentally remain optical quantities

that are compared to biogeochemical variables in models. Based on colour, the use of the blue/green band ratio algorithm does not only "see" chlorophyll, but also other type of in-water material. As such, reflectance-derived chlorophyll proves itself to be closer to the satellite retrieved chlorophyll than the chlorophyll computed in BAMHBI. Currently, ocean colour products are being assimilated in biogeochemical models that rarely resolve reflectance. This study hints at the opportunities provided by working directly with ocean colour derived from reflectances, that accounts for more than just chlorophyll and therefore may

be closer to the true meaning of such estimates. It also suggests that the assimilation of satellite surface reflectances has the potential to correct the IOPs and ultimately improve simulation of irradiance streams and reflectances.





## 6 Conclusion and perspectives

The modelling of light in the coupled physical-biogeochemical NEMO-BAMHBI framework for the Black Sea is enriched by adding a spectral radiative transfer model that explicitly solves the upward and downward light field at high spectral resolution.

This offers a direct connection between model variables and remote-sensed reflectance data, without relying on inversion algorithms. Four optically important water constituents are considered: pure water, phytoplankton, non-algal particles and CDOM. We aim at assessing the relative contribution of each constituent while accounting for the uncertainties in their IOPs using ensemble simulations. CDOM clearly dominates absorption at short wavelengths throughout the year while phytoplankton and non-algal particles dominate the (back-)scattering, having the largest influence during and following algal blooms.


As it is not explicitly simulated in BAMHBI, we assume higher uncertainty on CDOM IOPs than phytoplankton and non-algal particles. Ensemble show that perturbation of CDOM has a dominant influence on radiative fields. Its major role highlights the need for a better representation of this component in our biogeochemical model, which will require high quality data in the Black Sea. Such data are currently practically non-existent in the central areas and shelf zone, thus hampering modelling

capabilities.

The quality of the generated ensembles is assessed by comparing the ensemble distributions of irradiances, reflectances and reflectance-derived chlorophyll (rCHL) with observations. We find that the distributions generated by the ensemble do not capture all the spatial and temporal variability in measurements performed in the Black Sea. This is particularly the case

in coastal areas, and during winter and important blooms. This suggests that the introduction of uncertainties on IOPs is not enough to fully account for uncertainties in the RT model. First, vertical perturbations that could influence the position of the deep chlorophyll maximum are missing, and could be added to simulate uncertainties more realistically. Other sources need to be considered, such as forcings or parameterisation of the hydrodynamics and biogeochemistry. Their influence has been studied in other frameworks such as in Garnier et al. (2016).


Comparison to remote-sensed data shows promising agreement on reflectance ratios, even though reflectances themselves are less accurately modelled. Uncertainties on the atmospheric correction or interface effects (e.g. BRDF) are indeed removed using reflectance ratios between close wavebands. We also evidence that rCHL provide estimates of chlorophyll that are closer to remote-sensed data than the chlorophyll modelled in BAMHBI based on biological mechanisms. Using reflectances could

then allow to better characterise errors in the model.

Finally, in this paper, the coupling between the physical-biogeochemical model and the RT model is in one-way. It means that the chlorophyll and non-algal particles simulated by BAMHBI are used to compute the IOPs in the RT model but the light field produced by the RT model is not used to compute temperature and photosynthetic active radiation. Rather, a simple

one-stream optical model differentiating three wavelength is used to force the physical-biogeochemical model. We test a two-



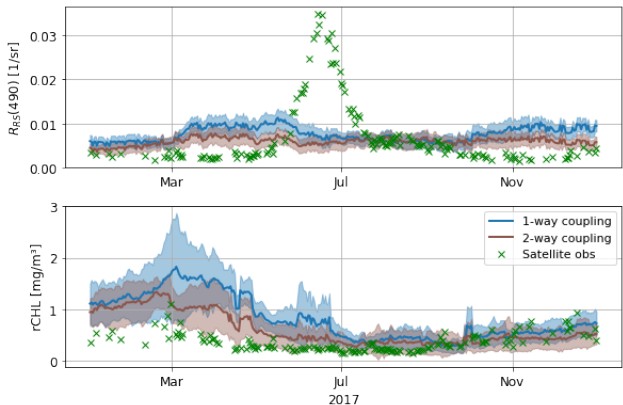

**Figure 15.** Time series of $R_{RS}(490)$ (left) and rCHL (right) at 43.22N, 36.63E with the 1-way (E4, in blue) and 2-way (in brown) configurations.

way configuration that, despite biases in the simulated radiative fields, shows promising results. The simulated time series of $R_{RS}(490)$ and rCHL are in better agreement with satellite data (Fig. 15). We find that in this 2-way configuration, the irradiance feedback towards the coupled framework does not disrupt the simulation of reflectance, although decreasing $R_{RS}(490)$ to bring it closer to satellite observations. For rCHL however, this first outlook shows that we are able to better compare remote-720    sensed data with this 2-way configuration, especially in the first months of the year. A two-way configuration for the coupled framework NEMO-BAMHBI-RT model will possibly pave the way towards the explicit assimilation of multi-spectral satellite reflectance in coupled physical-biogeochemical ocean models.

*Data availability.*    Ocean colour data was taken from Black Sea, Bio-Geo-Chemical, L3, daily Satellite Observations (1997-ongoing). E.U. Copernicus Marine Service Information (CMEMS). Marine Data Store (MDS). DOI: 10.48670/moi-00303 (Accessed on 05-03-2024). BGC-725    Argo data were collated within the Copernicus Marine Service (In Situ) and EMODnet collaboration framework. Data is made freely available by the Copernicus Marine Service and the programs that contribute to it. DOI: 10.13155/59938; 10.13155/43494 (Accessed on 06-12-2023).

## Appendix A:  Generation of $a_{cdom}$ forcing

CDOM is a major contributor to absorption in seawater, but is not explicitly simulated in the NEMO-BAMHBI framework. Therefore, we cannot use the same approach as for phytoplankton and non-algal particles. We choose here to derive a forcing 730    for $a_{cdom}$ from *in situ* BGC-Argo data. A collection of 625 profiles from floats 6901866 and 6903240 provide measurements of irradiance in 3 wavebands ($E_{Argo}$ at 380, 412 and 490 nm), chlorophyll concentration ($chl_{Argo}$), backscattering coefficient at 700 nm ($bbp_{700,Argo}$) and CDOM ($cdom_{Argo}$). Some profiles are discarded according to the quality flags of variables of interest here. Float 6901866 has measurements between June 2015 and June 2019 in the entire basin with limited coverage of



the central parts. Float 6903240 has measurements between April 2018 and July 2022, mostly around the Western Gyre. Both
floats do not cover the continental shelf, where dynamics of CDOM are more complex.

In the RT model, we parametrise the IOPs of phytoplankton and non-algal particles from biogeochemical variables of
BAMHBI. We use the same parameterisation to derive $a_{cdom}$. We choose a reference wavelength $\lambda_{ref} = 412$ nm in the range
of high CDOM absorption and with irradiance profiles measured by BGC-Argo floats to derive a reference $a_{cdom}$ profile. We
use the RT model to simulate the same profiles that are measured. IOPs of water are known in the same way as described
in Section 2.4.1. IOPs of phytoplankton are derived by using chlorophyll profiles from BGC-Argo data instead of BAMHBI
variables. However, we cannot discriminate between PFTs and therefore assume that chlorophyll is equally distributed between
the 3 PFTs solved in BAMHBI. IOPs of non-algal particles are derived from profiles of particulate backscattering coefficient
at 700 nm $bbp_{700,Argo}$. We assume this coefficient to be the sum of contributions from phytoplankton and non-algal particles
to derive a POC concentration that is consistent with our parameterisation:

$$POC_{Argo} = \frac{bbp_{700,Argo} - b_{b.phy}(700)}{b_{b,prt}^0(700)} \tag{A1}$$

We assume that the optical properties of CDOM on a water column are linear function of CDOM concentration, i.e. that on
a water column, all CDOM has the same absorbing power. We want $a_{cdom}$ profiles to have the same shape as CDOM profiles
measured by BGC-Argo float. We therefore assume:

$$a_{cdom} = k_{cdom}cdom_{Argo} \tag{A2}$$

with $k_{cdom}$ unknown. Starting with an initial value of $k = 1\,m^{-1}$, We then compute downwelling irradiance stream $E_{loop}(\lambda_{ref})$
for all profiles and compare them with BGC-Argo irradiance profiles. For each profile, the parameter $k$ is then updated to min-
imise the difference between $E_{loop}(\lambda_{ref})$ and $E_{Argo}(\lambda_{ref})$ profiles on the top 30m of the ocean (21 model layers), normalised
by their surface value for consistency in assessing the influence of IOPs. This simple optimisation problem is written:

$$min \sum_{k=1}^{21} \left( \frac{E_{loop}(\lambda_{ref})}{E_{loop}(\lambda_{ref})^{surf}} - \frac{E_{Argo}(\lambda_{ref})}{E_{Argo}(\lambda_{ref})^{surf}} \right) \tag{A3}$$

We use a deviation threshold of 2% to accept a k value for a given profile. Using this method for each profile of our collection,
we get a new collection of $k_{Argo}$ and then of $a_{cdom}(\lambda_{ref})$ profiles. We now store values of $a_{cdom}(\lambda_{ref})$ according to time and
seawater density as to link each time and location of the basin to a value for $a_{cdom}(412)$. We interpolate this dataset to generate
annual cycles for each density scale in order to smoothen the forcing. Fig. A1 shows the resulting forcing for $a_{cdom}(\lambda_{ref})$ for
selected seawater densities.



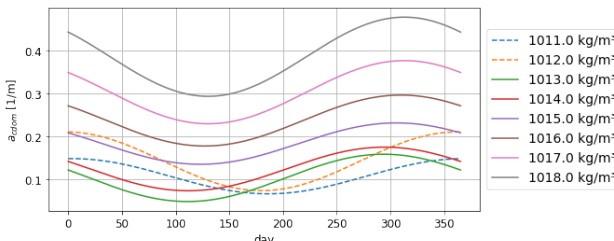

**Figure A1.** $a_{cdom}$ forcing scaled in density for $\lambda_{ref} = 412$ nm. This forcing was generated using a collection of BGC-Argo profiles in the deeper areas of the Black Sea.

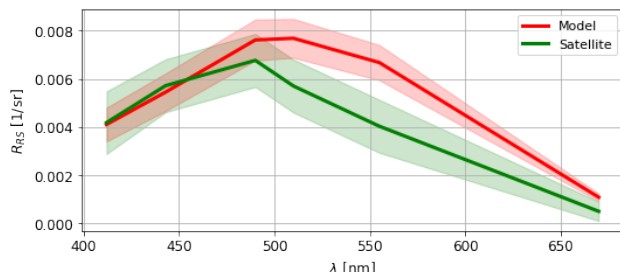

**Figure B1.** $R_{RS}$ average spectra in the eastern gyre over 2017. Uncertainties in model and remote-sensed data are represented in shaded areas.

## Appendix B: Model performance in the visible spectral range

In this paper, we focus on the wavebands that are used to derive rCHL, centred on 490 and 555 nm, respectively in the blue and the green. More generally, we computed irradiance and sea surface reflectance in the 6 wavebands that are found in the
multi-satellite reflectance product for the Black Sea made available by the Copernicus Marine Service, centred on 412, 443, 490, 510, 555 and 670 nm. B1 shows the average spectral sea surface reflectance in the eastern gyre of the Black Sea over 2017. As seen previously in the 490 and 555 nm centred wavebands, the model tends to overestimate sea surface reflectance. The model reflectance is consistent with remote-sensed data up to around 475 nm, after which the bias gradually increases for longer wavelengths. The deviation then decreases at the near infrared end of the spectral range.

*Author contributions.* LM, PB and MG conceptualised the research plan. LV completed the initial integration of the radiative transfer model into the NEMO-BAMHBI framework. LV and MG provided support with the NEMO-BAMHBI model. JMB and PB provided support with ensemble methods within the NEMO framework. LM performed the model calibration and ensemble simulations, analysis and wrote the first draft of the paper. All authors contributed to reviewing and editing the paper until its final version. MG provided funding through the BRIDGE-BS and NECCTON projects.



*Competing interests.* The authors declare that they have no conflict of interest.

*Acknowledgements.* This work was funded by the EU H2020 BRIDGE-BS project under grant agreement no. 101000240 and the EU HE NECCTON project under grant agreement no. 101081273. Insights from the ODESSA, MEDIATION and POSYDONIE projects were brought by the coauthors. Computational resources have been provided by the Consortium des Équipements de Calcul Intensif (CÉCI), funded by the Fonds de la Recherche Scientifique de Belgique (F.R.S.-FNRS) under Grant No. 2.5020.11 and by the Walloon Region. We

thank S. Dutkiewicz for sharing the MITGCM Darwin radiative transfer code. We also thank P. Lazzari, M. Baklouti, J. Lamouroux and P. Verezemskaya for helpful discussions.



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
