# Peer review of "Characterisation of uncertainties in an ocean radiative transfer model for the Black Sea through ensemble simulations"

_EGUsphere, 2024_

## Author Comment (AC1)

**Response to reviewer 1:**

*The authors investigate the influence of uncertainties in the inherent optical properties on the modelling of radiometric quantities adopting a radiative transfer model applied to the Black Sea. A detailed study is presented that considers several important optical components and discusses their role in light propagation. An advanced stochastic approach is used to understand the uncertainty and the relative contribution of plankton, nap, CDOM.*

*The results are clearly explained, the approach used is new and represents an important advance for biogeochemical modelling as it increases the realism of the models. Moreover, the analysis presented could be useful for the marine biogeochemical modelling community.*

*Below I list some minor revisions that I think could improve the text.*

We thank the reviewer for the careful review of our paper, positive comments and for the remarks and suggestions that will help us improve our paper and its clarity. We have detailed below our replies and updates according to these suggestions. Comments from the reviewer are in bold and our replies are in light below each comment.

*Pg2 lines 47-48 "Over the years, various models were developed with different approaches to refine the representation of light in models (Gregg and Casey, 2009; Mobley et al., 2009)." I would remove the repetition of models.*

*Pg 3 lines 69-70 "The penetration of the spectral irradiance is determined by the absorption and scattering properties of the medium that are derived from concentrations of optically active components, in 33 wavelengths." I would substitute with "The penetration of the spectral irradiance is determined in 33 wavelengths by the absorption and scattering properties of the medium that are derived from concentrations of optically active components."*

These has been changed according to the suggestions made by the reviewer.

*Pg 6 Eq.4 I would give the unit of measurement of PAR. Since the previous section mentions the one-way coupling with the RT model, I would explain why the PAR is introduced by the RT model since there is no feedback with biogeochemistry.*

The units of PAR have been added to the text. To justify the introduction of PAR despite the absence of feeback with biogeochemistry, the following sentence was added: "Although the PAR simulated by the RT model is not used in the coupled framework in a one-way configuration, it is a useful quantity that can be used for comparison with in situ data."

***Pg 15 Ensemble simulations. I would suggest that the authors create a table summarizing the properties of each of the 4 experiments.***

A new table was added summarising information on the 4 ensemble experiments: number of members and optical properties that are perturbed.

***Pg 17 line 426. "until in increases" should be "until it increases".***

Fixed.

***Pg 18 Figure 6 is very small and very difficult to read. Units of measurement in the y-axis are missing. In general, many of the illustrations in this manuscript are very small, especially the fonts, and the fonts, size and legibility should be enlarged.***

Units were added to figure 6. All figures except figure 1 have been remade with larger size and fonts to improve legibility.

***Pg 22 Figure 10, again the figure is very small. PAR should normally be expressed in quanta, and the PAR data from the BGC Argo float is also normally expressed in quanta. How did you convert to have W/m2? Perhaps the conversion formula could explain the discrepancy in PAR in Figure 10?***

Additional text was added to explain how the conversion is performed: "PAR data from BGC-Argo floats is given in units of µmol/s/m2. The conversion to W/m2, the unit used in model outputs, is performed using 1 W/m2 = 4.6 µmol/s/m2 to obtain comparable quantities. This approximation is given for daylight conditions in Thimijan and Heins (1983). In sunny conditions, this coefficient would normally have to be lower."

It is true that the use of a constant coefficient to perform unit conversion is an approximation. In particular, the conversion factor would likely be lower in summer, when sunny conditions are more frequent. It also happens to be the time of the year where discrpencies are the largest in irradiance and PAR data. However, if we refer to Thimijan and Heins (1983), the variations in the conversion factor would not be large enough to explained the discrepancies observed here. A sentence was added later in the text to highlight the potential role of the factor: "For PAR data, part of the discrepancies in summer could be explained by an inconsistent unit conversion in sunny conditions."

***Pg 22 lines 513- 521 Rank diagrams should be explained in more detail so that they are fully understandable, especially for the readers without expertise in ensemble modelling.***

This paragraph was rephrased to explain in more detail rank histograms: "To each observation is attributed a rank that is equal to its relative position the realisations of the ensemble for this observation. The entire set of observations is ranked within the sorted ensemble of corresponding

simulated data. A flat histogram evidences perfect reliability, i.e. an ensemble distribution that matches the distribution of observations. A convex rank histogram suggests that the ensemble is over-dispersive (all observations tend to be within the ensemble), while a concave rank histogram suggests that the ensemble is under-dispersive (observations tend to be outside of the ensemble). The extreme ranks correspond to observations that are lower or higher than all realisations of the ensemble."

*Pg 22 lin 524 iis the reference to Fig. 9 correct? The matching between model and data in Fig. 9 are very good, so the comment is unclear.*

The reference is correct, but the comment was indeed unclear. Figure 9 shows a good agreement in the upper ~15 metres for irradiance at 490 nm, but an overestimation below. The idea here is to comment on this overestimation deeper in the water column. The text was changed to: "Fig. 9 shows that despite the excellent agreement close to the surface (also evidenced in Fig. 10), irradiance tends to be slightly overestimated deeper in the water column, regardless of the perturbation. The ensemble is nonetheless close to representing the distribution of measured irradiance streams."

---

## Author Comment (AC2)

**Response to reviewer 2:**

*The authors use a biogeochemical ocean model coupled one-way to a 3 stream radiative transfer model to show how modelled chlorophyll based on modelled spectral reflectance band ratios are more consistent with both satellite ocean colour and biogeochemical Argo float data than chlorophyll estimates derived directly from their physical-biogeochemical coupled model system, NEMO-BAMHBI. They demonstrate that introducing uncertainties in the form of random perturbations of inherent optical properties of different water constituents, improves the simulated distributions of radiative fields. The study is focused only in the deep central areas of the Black Sea, due to the limitations of the CDOM forcing used in the experiment.*

*This is an excellent piece of work. It significantly advances our understanding of how inherent optical properties of water constituents can be used to constrain biogeochemical models, leading to improved modelled predictions. Moreover, it opens new possibilities for integrating biogeochemical modelling, in situ optical observations and ocean colour remote sensing. The authors also demonstrate that preliminary results from a two-way coupling test show even more promising results.*

*The paper is well organised and well thought out. The approach and methods are valid. Limitations and assumptions inherent in their approach are discussed in depth and perspectives for future work are provided. I would be happy to see this published in Biogeosciences. It is a very timely and welcome contribution which unifies in situ marine optics, satellite ocean colour and biogeochemical modelling communities.*

*I have listed some minor comments below for the authors to consider.*

We thank the reviewer for the careful review of our paper and for the remarks and suggestions that will help us improve our paper and its clarity. We also thank the reviewer for providing such a positive feedback. Please find below our replies to specific comments and updates we brought to the text. Comments from the reviewer are in bold and our replies are in light below each comment.

*All of the figures (except figure 1) should be bigger, with larger fonts on the labelling.*

All figures except figure 1 have been remade with larger size and fonts to improve legibility.

*A table summarizing the different reference and ensemble runs would be useful.*

A new table was added summarising information on the 4 ensemble experiments: number of members and optical properties that are perturbed.

***Some references are missing, or in the wrong place.***

- ***Line 36: add Cahill et al., 2008, https://doi.org/10.1029/2008GL033595***
- ***Line 49: add Bissett et al., 1999, https://doi.org/10.1016/S0967-0637(98)00063-6***
- ***Dutkiewicz et al., 2015 should be referenced earlier, I think, line 59 after the sentence "In recent years, … reflectance (Dutkiewicz et al., 2015)***

The suggested references were added to the text. The reference for Dutkiewicz et al. (2015) was also moved earlier as suggested.

***In places, the English should be improved for better understanding, e.g.***

- ***Line 39: "They can be described by the absorption and (back-) scattering spectra of each water …" instead of "They consist in absorption and …"***

Modified as suggested.

- ***Lines 67-68: at the end of the sentence "It solves the spectral wavebands corresponding to those typically used in remote sensing" add an example of what these wavebands are?***

The sentence was modified as: "Its spectral range includes the wavebands corresponding to those typically used in remote sensing (e.g. 412, 490 or 555 nm)."

- ***Lines 86 – 89: Suggest converting the following into a numbered list, easier to comprehend sequence of analysis. "We first assess the effect of … that are consistent with observations."***

This sequence was changed into a numbered list as suggested to improve legibility.

"This analysis comes in three steps:

1. Assessment of the effect of uncertainties in the parametrisation of absorption and (back-)scattering for each constituent separately.

2. Estimation of the combined effect of uncertainties for all uncertain constituents on irradiance and reflectance fields.

3. Evaluation of our ability to provide distributions of sea surface reflectance that are consistent with observations by modelling these uncertainties."

- ***Line 96: change last sentence to something like: "Finally in section 5, we discuss the limitations and assumptions of the study and provide an outlook for future work."***

- *Line 109: change to " … solved with NEMO 4.2 which is online coupled to the biogeochemical model."*
- *Line 143: change to "Attenuation coefficients are derived from absorption …".*

Modified as suggested.

- *Lines 169 – 172: this paragraph appears without any explanation, should be qualified with a statement which explains why the product is mentioned, e.g. to validate the modelled reflectances.*

Indeed a transition was lacking to explain why this product is mentioned. The text was changed to: "RRS ($\lambda$, 0+) is directly comparable to remote-sensed reflectance data and will more simply be noted RRS in the following. Among the available satellite products, we will be using here the multi-satellite product provided by the Copernicus Marine Service for the Black Sea for validation and comparison."

- *Lines 179-181: explain more clearly ecological reasons for choosing the band-ratio algorithm over the NN approach. Your study is focused on waters where the reflectance signal is dominated by phytoplankton, for example?*

The choice is motivated by our focus on the central parts of the basin were the band-ratio algorithm is dominant in the merged product (see Kajiyama et al., 2018). A sentence was added to explain this choice: "This choice over the neural network approach is motivated by our focus on the deep parts of the basin, where the band-ratio algorithm is predominantly used".

- *Lines 207 – 208: change to " … solved in BAMHBI: these are large flagellates ….and diatoms, all of which are the dominant species in the Black Sea"*
- *Line 249: change to "When run over the Black Sea, …."*
- *Line 260: change to "… water column, the water constituent IOPs would not be altered."*

Modified as suggested.

- *Lines 365 – 371: add table summarizing different simulations.*

Added as suggested, see comment n°2.

- *Line 426: change to " … until it increases again …"*
- *Line 430: change to "… at 555 nm, as its contribution to …"*

Fixed.

- ***Lines 447 - 448: change to "… phytoplankton early in the year with a lower contribution of CDOM in the ensemble spread."***
- ***Line 457: change to " … gradually increases with depth …"***

Modified as suggested.

- ***Line 562: "surface chlorophyll is defined as the average concentration over the top 10m." Is this based on some average of the 1st optical depth? Or? Maybe elaborate a little on this choice of depth over which to integrate the data.***

Yes, the depth of 10 metres corresponds to the average first optical depth in the Black Sea. We have added a reference and modified the text to support this assumption: ". For the BGC-Argo and BAMHBI deterministic run, surface chlorophyll is defined as the average concentration over the top 10 m, which corresponds to the average water optical depth in the Black sea (Peneva and Stips, 2005)."